# The Student Athlete Wellness Portal: Translating Student Athletes’ Prescription Opioid Use Narratives into a Targeted Public Health Intervention

**DOI:** 10.3390/medicina58111642

**Published:** 2022-11-14

**Authors:** Anne E. Pezalla, HyeJeong Choi, Francis McKee, Michelle Miller-Day, Michael Hecht

**Affiliations:** 1Psychology Department, Macalester College, St. Paul, MN 55105, USA; 2School of Health, University of Missouri, Columbia, MO 65211, USA; 3St. Andrews Development, York, PA 17401, USA; 4School of Communications, Communications Studies, Chapman University, Orange, CA 92705, USA; 5REAL Prevention LLC, Pennsylvania State University, State College, PA 16802, USA

**Keywords:** opioid misuse, technology, adolescents, athletes, public health, intervention, narratives

## Abstract

*Background and Objectives*: The opioid epidemic has permeated all strata of society over the last two decades, especially within the adolescent student athletic environment, a group particularly at risk and presenting their own challenges for science and practice. This paper (a) describes the development of a web-based intervention called the Student Athlete Wellness Portal that models effective opioid misuse resistance strategies and (b) details the findings of a single-group design to test its effectiveness. *Materials and Methods*: Formative research included 35 student athletes residing in the United States, ages 14 to 21, who had been injured in their school-based sport. They participated in in-depth qualitative interviews to explore narratives relating to their injuries and pain management plans. Inductive analyses of interview transcripts revealed themes of the challenges of being a student athlete, manageable vs. unmanageable pain, and ways to stay healthy. These themes were translated into prevention messages for the portal, which was then tested in a single-group design. *Results*: Users of the portal (*n* = 102) showed significant decreases in their willingness to misuse opioids and increases in their perceptions of opioid risks. *Conclusions*: This manuscript illuminates the processes involved in translating basic research knowledge into intervention scripts and reveals the promising effects of a technology-based wellness portal.

## 1. Introduction

Opioid use and misuse have migrated into the adolescent student athletic environment. This includes athletes who misuse their own prescription opioids (i.e., use them recreationally or for other non-prescribed purposes) as well as those who divert prescription opioids to others for their misuse [1,2]. It is estimated that 2% of youth athletes use heroin, although this, too, varies by sport [3]. Unfortunately, little has been conducted to address the needs of this at-risk population. This paper addresses these issues.

According to a recent study in the United States, the incidence of non-medical opioid use is higher among athletes than non-athletes [1]. This problem is not limited to the United States: statistics document the prevalence of non-medical prescription opioid use among athletes in European countries, as well, with some scholars [4], going so far as to reference opioid misuse as a “pandemic” when youth are injured [5,6]. The picture, however, is complex: some studies suggest that rates for athletes merely mirror those for the general adolescent student population [3], which still puts them at risk; some suggest an increased risk for male athletes but not female athletes [7]; and others suggest a greater likelihood of use among those participating in only some sports, such as hockey or varsity-level sports among male athletes [1]. Overall, it is estimated that as many as 14% of athletes misuse prescription opioids [8] and that prescription opioid misuse among athletes is associated with somewhat different risk factors from those associated with misuse in the general adolescent population [3], including the heightened perceived need to compete and arguably a higher pain tolerance.

Other reports have highlighted the problem of prescription opioid diversion (e.g., sharing or selling their opioids to others) among student athletes [9], and schools across the country have begun partnering with law enforcement for National Prescription Drug Take Back Day events to help students understand the risks of unused medications and prevent old medications from the diversion. These trends have underscored the importance of prevention interventions tailored to meet the needs of young athletes and keep them safe.

### Purpose

The overall goal of this paper is to describe in greater depth student athletes’ experiences with prescription opioids to provide the formative basis for intervention development and feasibility testing. Effective intervention requires a deeper understanding of the nature of prescription opioid misuse among high school athletes. While surveys reveal the breadth of the problem, they provide limited insight into the social and cultural framework of athletics within which misuse occurs, nor into the interpretive frameworks of the athletes themselves. Narrative research is one approach to providing a depth of understanding of the problem to develop engaging prevention interventions that resonate with athletes, especially those that are low awareness and/or resistant to prevention messages [10].

Accordingly, the specific purpose of the current research is to examine student athletes’ narratives and discuss the translation of those stories into a narrative-based prevention intervention for student athletes. In our formative research (phase one), we collected and analyzed athletes’ decision narratives around pain management, including the role of prescription opioids. In phase two, we translated those narratives into prevention messages for the narrative video intervention targeted to student athletes and titled the Student Athlete Wellness Portal.

#### Theoretical Framework: A Narrative-Based Approach to Health Promotion

“Sport,” wrote Kretchmar [11], “is story-friendly.” Sport naturally lends itself to a compelling storyline: the athlete who gets knocked down makes an inspiring comeback, the unlikely win from the underdog, even the growth that comes from a demoralizing loss—much of what occurs within a sport has action, meaning, and a storyteller’s arc. In considering the best way to learn from athletes and then reach athletes, we sought a framework that would capitalize on sport’s innate storytelling ability—narrative or storytelling.

Narrative Engagement Theory (NET) [12] gave us a means to that end, sharpening our focus on the meanings that student athletes ascribed to their sport, their injuries, and their pain management. Miller-Day and Hecht [12] define narrative as talk organized around significant or consequential experiences, with characters undertaking some action, within a context, with implicit or explicit beginning and end points and significance for the narrator or her or his audience (p. 2). Personal narratives are culturally grounded and have provided valuable insights into understanding when athletes play through pain and injury [13,14,15], how athletes interpret messages about nutrition [16], and how athletes internalize the need to detect and then self-report a concussion [17]. Past research has found that translating those insights into prevention messages featuring personal narratives can be engaging and effective in reaching youth and young adults with low awareness and/or high resistance [10,12] and provide modeling of behaviors enhancing self-efficacy [12]. Our work was guided by this narrative framework.

## 2. Materials and Methods

Despite the significance of the problem [3,9], the context of athletes’ decisions and behaviors related to the misuse and/or diversion of prescription opioids remains unclear. To illuminate these processes, we employed in-depth interviews to inform our efforts in building the intervention, including what topics to target and how to realistically model health decision-making.

### 2.1. Recruitment Procedures

Participants for the formative research interviews were recruited through seven American high schools, varying in size (large and small), demographics (rural and urban), and sports available within the schools. Coaches, athletic trainers, and administrators were contacted to request permission to interview their student athletes. Advertisements for interview participants were disseminated through Facebook and Instagram social networks and partnerships with community-based organizations. The ads requested the participation of student athletes ages 14–21 (an age range that was chosen to include athletes who were entering into and leaving high school, the prime demographic range for our intervention) to discuss their experiences with their sport, with injuries, and with pain management that involved prescription pain medication. For students under the age of 18, parental consent to participate was waived due to the tension we perceived between the expressed desire from athletes to talk to us and their simultaneous concern that their parents learn about their history of opioid misuse. Individuals who initially agreed to participate were asked to complete a brief screener survey, which confirmed participants’: (1) self-identification as a student athlete, (2) age of 14–21, (3) English speaking ability, (4) experience of an injury from their sport, and (5) willingness to participate in an interview.

### 2.2. Interview Procedures

All participants were interviewed by two professors who teach qualitative methods and two graduate students within their shared discipline; all have training in and experience with qualitative data collection and analysis. Due to COVID-19 and the geographic dispersal of participants, interviews were conducted via Zoom and were audio recorded and transcribed, with personal information removed. Interviews were open-ended and designed to explore student athletes’ experience with injuries and pain, experiences being offered substances (probing for prescription opioids and other opioids if not discussed), experiences with pressure to divert some of their prescribed opioids to others, any discussions of prescription opioids, and opioid use norms. The trained interview team explored the “who, what, when, where, why, and how” of conversations about opioids, generating narrative accounts of conversations where these substances were offered and the resistance strategies used. Reactions were elicited to several pre-existing drug abuse resistance interventions and websites. 

Each interview guide began with the following open-ended prompt, “We want to hear your stories! Please tell me about all the things you do to enhance your performance as an athlete.” Participants were then asked to reflect on the phrase “culture of pain” in their sport, on the role of prescription medications in sports, on other substance use in sports, and on two existing interventions targeting athletes in their age group: the ABCs of Prescription Drug Safety and a D.A.R.E.-sponsored module on opioids.

Participants were asked to share their experiences and assured that there were no right or wrong answers. To protect confidentiality, they were asked not to share full names, and no full names were linked to data files. Interviews ranged from 28 min (an outlier because of its brevity, which prompted a team-based meeting to brainstorm more follow-up questions to probe more deeply into athletes’ experiences) to 72 min in length. The average interview time was 52 min.

### 2.3. Analytic Procedures

Coding the formative interviews occurred in three stages. In the first stage, employing strategies of analytic induction [18], the lead researcher of this project (and one of the interviewers) provided the remaining three interviewers with a list of a priori themes based on the literature review: performance enhancement, a culture of pain, and prescription drug use. Our full interview team then read through a subset of transcripts to tag and label any additional codes that seemed relevant to the purpose of the study [19] and altogether created a codebook with a list of codes and their shared definitions. In the second stage, using this codebook, we divided into pairs. Each pair independently coded the same transcripts, then met shortly thereafter to compare codes. Only when paired coders reached an 80% level of inter-coder agreement (i.e., when at least 80% of their tagged transcripts contained the same codes) were we able to conduct line-by-line coding on our own. Once all transcripts were coded, we moved into the third and final stage of analysis: identifying themes across all transcripts and their relationship to one another to understand these student athletes’ experiences and identify any prototypical narratives for possible inclusion in the Student Athlete Wellness Portal (SAWP) intervention.

## 3. Results

### 3.1. Formative Research

A total of 35 student athletes (21 male; 14 female) participated in the interviews. Twenty-seven of the athletes played contact sports, as defined by the American Academy of Pediatrics [20]: football (*n* = 11), soccer (*n =* 7), hockey (*n =* 3), basketball (*n =* 3), wrestling (*n =* 2), and lacrosse (*n =* 1). The remaining student athletes played limited-contact or no-contact sports: volleyball (*n =* 4), baseball (*n =* 3), and gymnastics (*n =* 1). Participants ranged from 14–21 years of age, 60% Caucasian (*n =* 21), 31% African American (*n =* 11), and 9% Asian (*n =* 3).

Guided by the literature and by our ongoing interrogation of and discussion about the data, we identified the following themes as the most salient within our analysis of student athletes’ narratives about their sport and pain: (1) the challenges of being a student athlete, (2) manageable vs. unmanageable pain, and (3) ways to stay healthy. These themes reflect participants’ informal and formal influences that shape their awareness and attitudes about wellness and about taking opioids to manage their pain. The names used below are fictional.

#### 3.1.1. The Challenges of Being a Student Athlete

##### Stressful Duality of Student/Athlete Roles

Participants’ stories revealed a tremendous amount of stress. They were expected to excel not only in their sport(s) but also in their academics, a dual expectation that left them feeling overwhelmed. Mike told us, “*You have to be on point in school and you have to be on point in football. Every day, it’s like okay, I have a test or I have to study or I have meetings or I have practice or I have a game. It never ends*.” The rigor of both the athletic practice and their schoolwork demands left these athletes with literally no time to spend on any other stress-relieving activities other than eating and sleeping.

These expectations were particularly high given the challenges of the COVID-19 pandemic, where the number of practices or games were curtailed due to sickness or quarantine restrictions, so when there was a practice or game, coaches were adamant about players “giving it their all.” Abby explained that “*I was pushed by the coach to return as soon as my [COVID-19] test came back negative, and then it was go, go, go. It was really manipulative and bad. So, I was pushed back way too hard and way too early than what I should have been, and then it was partially my fault too, because I knew it was going to be my last game. So, I was like, ‘I’m playing. I don’t know how; I don’t care what it looks like. I just want to play.’*”

Others felt the pressure of performing in their sport because they were nearing their high school graduation and knew their opportunities for playing college-level sports were being considered by outside recruiters. Christine recalled a time when her broken thumb was disregarded as simply being “jammed” and was forced to play for a high-pressure game when a recruiter was watching her and her team: *“It was a showcase thing for colleges. And my thumb got really swollen, so I had the trainer look at it. But he said, ‘oh, it’s not broken. It’s just jammed.’ So, they never gave me the opportunity to sit. I just thought I was obligated to keep playing. And then I got it checked out a few weeks later and discovered that I had in fact broken my thumb.*”

These expectations to play and to perform, sometimes at the cost of the athletes’ physical or mental wellness, often exacerbated their pre-existing injuries or directly contributed to a new injury.

##### Culture of Pain

Many student athletes spoke plainly of the expectation that they would get hurt in their sport, illustrating what was coined as *the culture of pain*. *“It’s not if you’ll get hurt*,” said John, a football player; “*it’s when*,” suggesting that injuries are inevitable in the sport. From Mike: *“It’s like you’re guaranteed to have injuries*.”

Some athletes spoke of the merits of pain in their sport, noting that getting “knocked down” can teach you about overall hardships in life and how to handle them. “*I think it [getting injured] has made me stronger in some ways, it’s built up my confidence*,” said Lincoln. Almost every participant spoke about his or her high pain tolerance, suggesting that they were not discouraged by the pain associated with their sport but were often energized by it. “*I mean, I personally loved it* [the physicality of sport]”, said Natalia, a lacrosse player. “*I’ve liked being aggressive. I liked it, and I liked when I got hurt. It just made me even more mad and fueled up to work harder.*” Harris spoke of the “adrenaline rush” of tackle football, and how, when he was injured and could not play until he was healed, he slept poorly because he missed the excitement of his contact sport: “[After my injury], I *wasn’t able to sleep, not because of the pain, but because I’d wake up in the middle of a dream where I was playing football. And my whole body would be shaking. And that made me think that most of the reason people play these sports is because of the adrenaline rush you get in the moment. And I wasn’t able to get that after my injury*.”

Almost every participant had more than one injury from their sport: broken legs, toes, fingers, arms, and concussions. The most common injuries, though, were injuries to the shoulder, typically a torn labrum, and injuries to the knee, typically a torn ligament, most requiring reparative surgery and post-operative sessions with a physical trainer.

#### 3.1.2. Manageable vs. Unmanageable Pain

We asked participants, “Under what circumstances does pain become unmanageable? Tell us a story.” Participants would sometimes muse that unmanageable pain was pain that kept them awake at night. Some said that unmanageable pain was pain that did not allow them to stand or sit comfortably. Others, though, gave a response that was more about how well they could perform in their sport rather than how pain-free their body was. Christine explained: “*I can handle pain. I have a high pain tolerance. But I guess my pain became unmanageable when it got to a point where my swing was just not as powerful as it had been. I could feel it wasn’t as strong as before*.” Soccer player Trinity shared a similar sentiment, noting that she felt her pain had become unmanageable when she was not able to run as fast as she wanted on the field.

We asked all participants, “Under what circumstances, if any, do you think it’s okay to use prescription opioid medications?” Their answers were universal: When a doctor prescribes them to you. That widespread belief, which is in alignment with the practices of the American Academy of Pediatrics, was sometimes contradicted by participants’ actual behavior. Two participants, Joey and Mike, took their prescription opioids for longer than prescribed—Joey for three months, Mike for several years—but the remaining participants stopped taking their opioids far earlier than their doctors had prescribed, preferring to “feel the pain” instead or simply deciding that the effects of the opioids were worse than the pain from their injury or healing. Indeed, with the exception of Mike, who had an almost euphoric experience after taking prescription opioids in the hours after an invasive hip surgery (“*it was like, lights out. Like, I can do this thing called life now!”*), every other participant shared stories of unpleasant side effects of their prescribed opioids. “*I felt like a zombie,”* Makayla said, “*I couldn’t focus at all.*” Anthony and Josh both shared that they had terrible gastrointestinal pains and were “*totally blocked up*” from their medication, needing stool softeners to ease their discomfort. Some athletes took the obligatory dosage of their medication after their surgery, but once they were home from the hospital or clinic or pharmacy, they discontinued using their prescription medication altogether, relying instead on Tylenol or Ibuprofen or simply forgoing pain medication completely.

#### 3.1.3. Ways to Stay Healthy

All the athletes we interviewed expressed an earnest desire to keep themselves strong, well-nourished, and injury-free for their sport. We organized these sentiments into a fairly broad category that we called “Ways to Stay Healthy” and then identified two distinct sub-themes within it that merited inclusion in the intervention due to their frequent mention: My Mother as my Ally, which positioned athletes’ mothers as important gatekeepers of their prescription opioids; and My Body as a Temple, which illustrated the ways that athletes monitored what they ingested into their bodies, including food and medicine.

##### My Mother as My Ally

Out of the 35 athletes we interviewed, 15 referenced the role their mothers played in helping them avoid opioid misuse. Here are several elicitations from participants, each illustrating the theme we called “My Mother as my Ally”:

Danielle:
*“I depend on my mom to take care of the details [of my prescription].”*


Chris:
*“My mom counts the pills so I know I can’t abuse them.”*


Abby:
*“My mom would freak if she found out I snuck any [of my medication].”*


Joey:
*“My mom is in charge of the pills, so I just take what she gives me.”*


These elicitations may suggest that the athletes are controlled by their mothers, but this sentiment was not shared by any of the athletes we interviewed. Rather, they were comforted by the fact that their mothers “took care of” the storage, allotment, and disposal of the drugs. To the athletes, their mothers’ active role in managing their prescription medicine relinquished them of the stress of handling the medication on their own.

##### My Body as a Temple

Twenty of the athletes we interviewed explicitly mentioned how dedicated they were to taking care of their bodies, a theme we termed “my body as a temple.” Here are several illustrative excerpts:

Mike:
*“I was on a 6000 calorie diet a day in order to maintain my body weight and gain weight, and that was me recording every single thing I’ve put into my body. And it’s very tough to get to 6000 friggin’ calories. And then I was meeting with a nutritionist at the end of the week to go through everything I put in my body.”*


Mikayla:
*“I’m really cautious about what I put in my body.”*


Hilton:
*“Call me a weirdo but I want to feel pain if I’m injured. I don’t want to dull it.”*


Harrison:
*“I know I have an addictive personality, so I’m freaked out about opioids. I think I’d like them too much, so I’ve stayed away.”*


### 3.2. Summary

These three categories—on the challenges of being a student athlete, manageable vs. unmanageable pain, and ways to stay healthy—provided us with a foundational understanding of how student athletes manage their pain and stay healthy. Within those three categories, we found powerful narratives of athletes showing remarkable discipline in their training and close reliance on trainers, nutritionists, and other health professionals to engage in stretching and cross-training and mindful eating. This information provided us with content for the intervention.

### 3.3. Intervention Development

Interventions targeting student athletes are needed to address the potential misuse of prescription opioid medications [1]. A targeted intervention seeks to influence a specific population or group as opposed to the overall population or at an individual level [21]. With evidence to suggest that student athletes are at a higher risk for prescription opioid misuse in comparison to the general adolescent population [1,4,5], the second phase of our research focused on developing a narrative intervention specifically for high school student athletes. To do this, we set out to translate the themes from the athletes’ narratives in our formative research into prevention messages specifically targeting young athletes.

#### 3.3.1. Scripting Narratives for the “Student Athlete Wellness Portal” Intervention

Presenting health information in a manner that is clear and comprehensible is central to the effectiveness of health promotion initiatives [22]. As discussed earlier, creating messages that use narrative, or are story-based, provides information in a story with characters, settings, and actions that are accessible, easy to understand, can target beliefs and norms, and become a vehicle to promote health behavior change. To accomplish this, salient findings from the formative interviews must be translated into clear, convincing, “prototypical narratives” that can promote engagement among the target demographic (student athletes). A prototypical narrative is a usual or quintessential representation of the collected narratives, reflecting the structural as well as content dimensions of participants’ experiences [14].

The structural elements that cut across the interviews included pain management discussions with their parents, teammates, friends, healthcare professionals, and media messages. Contexts and settings included the doctor’s office or athletic trainer’s room, the locker room, or in informal situations (conversations while hanging out, gaming, and socializing). Our interpretation of the data suggested three different scenarios, which we selected based on their potential for engagement and for relevance among student athletes, to be scripted for the intervention. An example of this process is outlined in Table 1.

#### 3.3.2. Scripting the Intervention Videos

The general rule for scripting that our team has developed over the past decade of scripting these kinds of health messages is to follow these general guidelines: (a) select a prototypical narrative to tell that has a clear point of view and a distinct beginning, middle, and end, (b) define the details of the message including length, tone, key message concepts, (c) weave together health facts within the context of the narrative account, (d) provide positive modeling of the health behavior, (e) include a call to action, (f) address benefits of the action, and (g) enhance viewer’s agency (citation removed for blind review).

Once initial scripts were drafted, they were reviewed by five student athletes ages 19 to 22 and one athletic trainer, identified because of their expressed interest in the project and their voluntary willingness to review a prototype to obtain feedback on the realism, accuracy, and interest in the scripts. Minor revisions were made based on the feedback. These initial scripts were then sent to Academy Award-winning filmmaker Aaron Matthews at Look Alive films, who collaborated with a screenwriter to refine the scripts and have them formatted and ready for casting and production. The final scripts were the result of a collaboration among many professionals but grounded in the experiences of the student athletes in our formative interviews.

#### 3.3.3. Creating the Student Athlete Wellness Portal

While the filmmaker and screenwriter created the videos, we were simultaneously putting together the full Student Athlete Wellness Portal, which would include the videos. The SAWP curriculum adhered to a VODEPS pedagogy, with our goals to (V) visually engage the learner, (O) provide an overview of the level content, (D) define terms and learning objectives, (E) demonstrate and provide examples, (P) practice skills, and (S) summarize the level. To that end, we prioritized high-resolution images and voiceovers of athletes; clear and bold headings for the three main modules; ample opportunity for interactive practice, including drag-and-drop and slider options in response to questions; and summaries of each module. It was important to keep the scope of the portal within limits, too, so we made adjustments to keep the portal approximately 10 min long for users in order to maintain high engagement.

SAWP programming consisted of client-side technologies employing HTML, CSS, and JavaScript code sets. The server-side technologies utilized Microsoft.Net, Web Services, and Microsoft SQL Server database. Approximately 27,000 lines of code were created and utilized. Additionally, the after-deployment and analytical phase required substantial extraction and presentation efforts, all resulting in a precise data set utilized by the team to determine program efficacy.

### 3.4. Discussion of Development

From the student athletes in this study, we learned about the stressors of the student-athlete dual roles, the reevaluation of “manageable” pain, and athletes’ awareness of how to take care of their bodies. Our intent in describing these details was two-fold: to contribute to the scholarly literature on pain management among student athletes and to demystify the process of employing a theory-informed message design for narrative interventions. Next, we turn to the feasibility trial to discuss the procedure and share our results on the effectiveness of the SAWP among our target population: student athletes.

### 3.5. Feasibility Trial

The first step in exploring the preliminary evidence of the usability, engagement, and program outcomes of the SAWP was to conduct a feasibility trial. Because of the preliminary nature of this exploration and with precedent from other comparable studies on athletes and wellness, e.g., [23], a single-group, pretest–posttest design was used.

#### 3.5.1. Procedure and Participants

High school student athletes were recruited across states (e.g., TX, NJ, PA, and others) to participate in the feasibility study for SAWP. Schools with established athletic programs were identified, and contact was made at a high level, i.e., the principal or athletic director. After describing the purpose and process of the program, the concept was advanced to the board or superintendent level, and approval to proceed was secured.

We pilot-tested the procedures on a small group of athletes. Parental permission was secured by completing a paper form and entering a unique identifier into the program to proceed. The challenge posed by the COVID environment and the start–stop nature of athletics during this time presented enormous challenges to the team. Few athletes were recruited and upwards of 60% of users who began using the program failed to finish it.

The initial approach of face-to-face recruiting gave way to the creation of an engaging, interactive recruiting model where custom cards containing a program URL and a QR code, each of which would direct the user to the SAWP program. This, along with some social media and direct contact with a new batch of schools, contributed to its ultimate recruiting success. In addition, the program was shortened (the introduction was made about two minutes shorter by omitting what was deemed to be a superfluous video); further, parental permission was integrated into the utilization process where a handoff from parent to child was made, resulting in a streamlined, automated consent workflow process.

Parental consent and youth assent were obtained for those 18 and younger, with only youth consent for those aged 19. Individual survey links were sent to participants through email or text. Self-reported assessments were collected at baseline (T1) and an immediate posttest after intervention (T2).

#### 3.5.2. Measures

Usability (immediate posttest) was measured with 10 items adapted from the SUS scale [24] with ratings on an agree-disagree scale of 1–5. Higher scores were coded to reflect greater usability. The reliability of this usability scale was acceptable (Cronbach alpha = 0.81).

Engagement (immediate posttest) was measured with eight items adapted from the Narrative Engagement Scale with ratings on a five-step agree-disagree scale. Higher scores were coded to reflect greater engagement. The scale measures three sub-constructs: interest, realism, and identification. Three items were used to measure interest (Cronbach’s α = 0.58) and realism (Cronbach’s α = 0.45), respectively. Two items were used to measure identification (Cronbach’s α = 0.60). These reliabilities were less than desirable, and we note that statistical tests of these data should be interpreted with caution.

Self-efficacy (pre and posttest) was measured by two items adapted from previous research on the self-efficacy to resist substance use [25] with ratings on a five-step agree-disagree scale [26]. Two items were averaged, with the higher score indicating higher efficacy (Cronbach’s α = 0.73).

Willingness to misuse (pre and posttest) was measured by two items developed for this study assessing how athletes respond to sports injuries and unused opioids. SAWP presented two scenarios focused on these issues (e.g., a situation that a student athlete has pain due to a sports injury and has opioids leftover), so we were able to compare how willingness to misuse at baseline was different from one in immediate follow-up based on the effects of SAWP. Two items were used, each presenting a scenario and providing four potential responses to the scenario. Participants checked which they would be willing to do, only one of which represented being unwilling to misuse and the other three indicating misuse. The items were scored 1/0 to reflect a willingness to misuse or not and calculated the percentage willing to misuse across both items.

Perceptions of opioids (immediate posttest) were measured by a single item at posttest indicating the degree to which participants perceived regular use of opioids as risky rated on a four-point scale from “No Risk” to “Great Risk”.

### 3.6. Results

The final sample consisted of a total of 102 athletes (49 female and 53 male) with an average age of 16.6 years (SD = 1.50, range = 13–19). Of these, the majority described themselves as White (*n* = 57, 55.9%), followed by Black or African American (*n* = 17, 16.7%), others (*n* = 10, 9.8%), Hispanic (*n* = 8, 7.8%); Asian (*n* = 5, 4.9%), and American Indian/Alaska Native (*n* = 5, 4.9%). Participant sports included football (*n* = 31, 30.7%), baseball/softball (*n* = 22, 21.8%), soccer (*n* = 20, 19.8%), volleyball (*n* = 22, 21.8%), wrestling (*n* = 16, 15.8%), gymnastics (*n* = 11, 10.9%), lacrosse (*n* = 9, 8.9%), ice hockey (*n* = 5, 4.9%), and cheerleading (*n* = 2, 2.00%). Seventy-seven participants (75.5%) reported that they had lifetime sport-related injuries and 44% (*n* = 44) reported lifetime opioid misuse, and 28% (*n* = 27) reported past 30-days opioid misuse.

Lacking a control group, statistical tests examined whether scores were higher than neutral for usability and engagement, which appear only on the posttest, and compared pretest and posttest scores for efficacy and willingness to misuse, which appear on both pretest and posttest. Perceptions of opioids were measured only in the posttest due to the limit of survey length.

Usability: We tested whether the mean of usability was higher than the neutral point 3 on the posttest using a one-sample *t*-test. The result showed that the average usability score (*M* = 3.49, *SD* = 0.65) was significantly higher than 3, *t* (90) = 7.28, *p* < 0.001.

Engagement: We also tested whether participants’ engagement with SAWP was higher than the neutral value of 3 on the posttest using a one-sample *t*-test. The result showed that the average interest score (*M* = 3.50, *SD* = 0.72) was a significantly higher score than 3, *t*(93) = 6.69, *p* < 0.001. The realism score (*M* = 3.47, *SD* = 0.73) was significantly higher than 3, *t*(93) = 6.19, *p* < 0.001. However, the identification score (*M* = 3.12, *SD* = 0.87) was not significantly higher than 3, *t*(93) = 1.31, *p* = 0.19.

Self-efficacy: We compared self-efficacy scores at baseline and at the immediate posttest. Using a paired *t*-test, we found that there was no significant difference between baseline (*M* = 4.01, *SD* = 1.05) and follow-up (*M* = 3.91. *SD* = 1.02), *t*(86) = −0.08, *p* = 0.38).

Willingness to misuse: We compared how willingness to misuse at baseline was different from ratings at the immediate follow-up and found that a significantly lower percentage reported willingness to misuse after the curriculum (*x*(1) = 52.07, *p* < 0.001). At baseline, approximately the same number of participants indicated a willingness to misuse (*n* = 44; 50.6%) as an unwillingness to misuse (*n* = 43; 49.4%). After SAWP at the posttest, 15% of students (*n* = 7) who reported willingness to misuse at baseline were unwilling to misuse. Another way of looking at this is that among the athletes, 43 indicated no willingness to misuse at baseline, and 40 (93%) remained as unwilling to misuse the opioid.

Perceptions of opioids: We tested whether perceptions of opioid risk score were higher than the mid-point, 2.5 (scale range: 1 (no risk)-4 (great risk)) on the posttest using a one-sample *t*-test. We found a significant difference, with participants perceiving “opioid misuse” as a moderate risk (*M* = 2.93, *SD* = 1.03, *t*(89) = 4.01, *p* < 0.001).

### 3.7. Discussion for Feasibility Study

The feasibility study demonstrated that SAWP has adequate usability and engagement, with both exhibiting scores above neutral, as well as promising effects on willingness to misuse and perceptions of opioid risks. No significant differences were found in self-efficacy. While the latter finding is less desirable, it is understandable that self-reports were provided on a posttest immediately following participation in SAWP that did not allow time to practice skills and test learning. Overall, however, these findings suggest that SAWP needs some further development to increase impact, including adding features to improve engagement and more attention to skill development, but it shows promise for reducing opioid misuse among high school athletes.

These conclusions must be tempered by several limitations. First, the COVID-19 epidemic truncated the feasibility study to a single-group design. Without a control group, firm conclusions cannot be reached about program outcomes. Second, the low reliability of the engagement scale should be carefully considered, although the scores were significantly higher than the neutral ones. Possibly, lower reliability was due to the time constraints of the overall project. Because students completed the pretest, went immediately into the intervention, and then transitioned to the posttest, they may have confounded the engagement and usability of SAWP with the pretest experience.

## 4. Conclusions

This paper reports on a case study of the development and feasibility testing of a narrative intervention to reduce prescription opioid misuse among high school athletes. The Student Athlete Wellness Portal, or SAWP, was developed iteratively based on the Narrative Engagement Theory using formative interviews to develop program content. The paper describes this process in detail, something rarely found in the prevention literature despite the emergence of interest in narrative interventions due to their demonstrated ability to engage low awareness and/or resistant audiences [10]. This paper portrays how to move from these stories to an intervention that proved usable and somewhat engaging, with promising effects on targeted outcomes. Given that our sample had a higher percentage of misuse compared with other studies [5], these findings hold promise.

We are considering a couple of improvements given our feasibility trial. First, we need to improve the level of student engagement. Given that we found a lower score on identification, we might need to consider what exactly made adolescent participants feel less connected to the actors in the SAWP; another qualitative study might be a good way to find out the reasons. Moreover, the difference in self-efficacy scores between baseline and follow-up was not significant. Given that it requires time to practice these skills, we suggest providing an immediate survey one month after the intervention. Furthermore, it is important to test whether participants’ actual behavior (i.e., misuse) changed over time. We could not test the behavior change due to the limitation of the study design. Yet future studies should investigate behavioral change. We speculated that most adolescents used smartphones to complete the intervention and surveys; thus, the online training and surveys should be optimized for mobile apps rather than a computer-based platform [27]. The interface between the two could be different, and it can substantially influence the outcome of the program [27]. Therefore, the intervention with technology should be carefully designed for the targeted population. Considering the rapid advancements in technology, we are hopeful for continual improvement and refinement in the delivery of these important interventions for student athletes.

## Figures and Tables

**Table 1 medicina-58-01642-t001:** “Student Athlete Wellness Portal” Video Intervention Example.

Title Video 1	Telemedicine
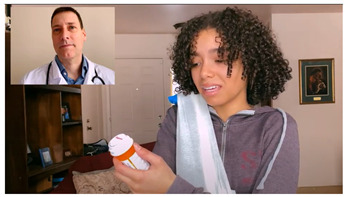	
Narrative Storyline	Talking about Opioids with your Doctor
Message Length Message Tone Message Key Concepts	2 min Conversational, informative Severity of opioid medication side effects; the benefits of talking to healthcare providers; acknowledgment about the discomfort in healing from an injury; reassurance that it will get better.
Health Facts to Include	Benefits of opioids, adverse side effects of opioids, and correct dosage or opioids.
Modeling	Questions to ask; Encouragement to talk with your healthcare provider
Call to Action	Questions about your meds? Talk to your pharmacist. Talk to your doctor. No question is a dumb question.
Benefits of Action	AVOID prescription medication misuse. Your body is a temple.
Enhance Agency	Verbal reassurance. Brevity. Humor. Tag lines

## Data Availability

Please contact HyeJeong Choi at choihyej@health.missouri.edu for access to the analyzed data.

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
