# Peer review of "The Student Athlete Wellness Portal: Translating Student Athletes’ Prescription Opioid Use Narratives into a Targeted Public Health Intervention"

_medicina, 2022, doi:10.3390/medicina58111642_

Round 1

Reviewer 1 Report (Previous Reviewer 1)

Interesting to read the new resubmitted article and think it is improved. However, I have som questions/conserns. 

In line 69-71 you write: "the specific purpose of the current research is to examine student athletes’ narratives and discuss the translation of those stories into a narrative-based prevention intervention for student athletes". And, in line 93-96 you write: "Using a narrative framework to guide our work, our goal was to shed light on the larger social and cultural framework of student athletes’ wellness, and then to re-story their narratives into a wellness program that would resonate with other athletes, with the potential to shape their attitudes, intentions, and behaviors". What is the aimof this study, as these two goals are not the same, understanding students' narratives and to shed light over the larger social and cultural framework seems to be two different aspects. 

Line 248-249, and line 251-253: whay are they in bold italic?

Line 263, what is ACL (torn ACL)?

Manageable vs. Unmanageable Pain: Both quotes in this paragraph can be interpreted as saying more about psychological pain than physical pain (did not feel it wasn’t as strong as before, and wasn’t able 274 to run as fast as she wanted). Does this study make any distinction between physical and psychological pain, the cause of the pain? Perhaps you should say something about this at the introduction section?

I can see you have improved the section "My mother as an ally". However I am still not convinced that the section about "ways to stay healthy" and especially the part that deals with "my body as an temple" gives us so much in-depth insight and how this relates to pain and the aim(s) of this study.  I think the two first qoutes doesen't fit. 

"My mother as an ally"  -the qoutes need a corresponding identity name

Line 326-328. I do agree in you analysis here, however, how is this related to pain management?  Please expand/explain this relationship a bit more.

Line 659 and forward: delete these lines. 

Author Response

Reviewer 2 Report (Previous Reviewer 2)

None

Author Response

No response needed. 

This manuscript is a resubmission of an earlier submission. The following is a list of the peer review reports and author responses from that submission.

Round 1

Reviewer 1 Report

Review: The Student Athlete Wellness Portal: Translating Student Athletes’ Prescription Opioid Use Narratives into a Targeted Public Health Intervention.

Thank you for the opportunity to read this paper. The topic is interesting, and the paper I well written. However, there is some weaknesses and unclearness with the current study that I will try to highlight.

Introduction:

·         Where is this study conducted (Europe/Asia/Africa/America)? It would be nice to know the context.

·         In general, there is a lack of literature on the extent of use of pain relievers/prescription opioids in sports internationally for this age group. . In addition, I lack background literature related to interventions for youth/young adults. Such information will strengthen this article and provide increased insight and the papers discussion would be improved.

·         Page 1, line 35, the authors write: "according to a recent national study", which nation?

·         Page 2, line 43, “in only some sports” – can you give examples of the type of sport this applies to?

·         Page 2, line 86: is this the correct page reference in the article to Miller-Day & Hecht?

·         Inadequate explanation of the rationale for how the testing of single-group design

Methods:

·         There is a lack of information about how the participants were selected for interview - why these 35 athletes? On what basis were they selected, and not just stating they were selected due to meeting the inclusion criteria.

·         Who did the interviews, and what kind of interview training did they have?

·         What was the background for having a screening survey when the participants had to meet the inclusion criteria to participate? It seems unnecessary to have a screening survey just to determine the inclusion/exclusion criteria.

·         There is no information about consent to an interview, how was that handled?

·         Are children under the age of 18 competent to consent, or do parents have to consent to children's study participation? This issue is not discussed in the article.

·         Page 3, line 120-127: this is results and ought to be moved to the result section.

·         Page 3, line 135: who is "the trained interview team"?

·         According to the paper, interviews lasted from 28 to 72 minutes. What was the average time for the interviews? Moreover, can one get in-depth information in a 28 min interview? Bringing this into the discussion as an ethical consideration will strengthen the article.

·         To be able to say something about this study's validity, there is a need for more information about the analysis process. I find it a bit superficial and would like more information here:

o   Was it only one person who did all the coding? (See page 4, line 153).

o   What background did this person have (and the team as well), and how did this background influence the coders perspective/views? Ethical and reflective considerations here would be nice.

o   How did you ensure an 80% inter-code agreement? (see page 4, line164)

o   Who participated in the team (page 4, line 163)? Only the writers or did other participate too?

o   How did you identify themes?

Results:

·         Page 4, line 171: “the following themes emerged….” here you might present all the topics presented in the result section.

·         Can the themes emerge? In an analysis process, the themes do not "appear by themselves", but there are codes/themes that the researchers themselves choose - this is a choice based on preferences, knowledge etc. This is also something that can be dealt with in the method section, and which will improve the article.

·         Nice to use quotations in the text, but it would have been better if they were in italics so that they were more visible - and be careful that you use the "-" sign before and after all the quotations. Missing quotation marks on page 5.

·         Page 6, line 250, «Percocet» - why this particular medication? Perhaps an explanation of which preparation this is, as it might not be internationally known.

·         Ways to stay healthy, page 6 line 259-280. This part needs more processing. It is not intuitive how these quotes illustrate the themes. The quote on page 6, line 265, how does this illustrate that the mother is an ally when the mother apparently has control over the number of pills?

·         Summary, line 282-285: I am not sure if the topics (challenges of being an athlete, manageable vs. unmanageable pain, and ways to stay healthy) gives us so much in-depth insight into how they manage/handle their pain and stay healthy. However, the results says something about how they experience the pressure of expectations, how pain is experienced and how the pain affects their life. Do you have more data on what the participants specifically do to manage the pain (besides ending prescribed use earlier or possibly go beyond the prescribed dose/period)? Do they train alternatively/other forms of training? What do they do when they cannot sleep due to the pain? How do the young people/young adults understand being healthy in a context, you are studying? Alternatively, how do they stay healthy when using opioids beyond the prescribed dose/time? Because this says something about what they actually do and may increase our knowledge. As I see it, answers to such questions will provide more in-depth understanding about your study first aim.

·         Page 8, line 324-325, “five student athletes aged 19 to 22 and one athletic trainer” – how are these participants selected, and why them? Is it completely random or are there some underlying selection criteria?  I miss an explanation of these aspects.

·         Page 8, line 352, I am not sure if you have enough coverage in the data to say it's a “hyper-awareness” of how to take care of their body. If so, it should be better substantiated.

·         Page 9, line 359: The abbreviation (SAWP) should appear earlier in the article, first used in line 71

·         Page 9, line 377: “In addition, the program was shortened” - why, and what was removed?

·         Page 9m line 385-394: This is results, and should be moved to the result section.

·         Page 9, line 400-401: “higher score were coded to reflect greater engagement”. How is it possible to gain a higher score than 5 on a 5-step agree-disagree scale?

·         Immediate posttest, what does this mean? Is the test taken directly after the test, within a few hours after or on the same day/week as the test was completed?

·         Page 10, line 406: “pre and posttest”, what is the time window for the pre and posttests?

·         Page 10, line 406: “two item adapted from previous research” - which item this is and why are they included? A reference to previous research where this is mentioned must be included.

·         Page 20, line 414: “ORRS” - what is this abbreviation?

·         It is a bit unclear why you present different scores as to whether they were lower or higher than the mean without this being linked to relevance for the intervention. What are you actually measuring and what do you want to show, it is a bit unclear for me.

Discussion:

·         As a reader, I cannot see that this is a discussion; it is more a form of a brief presentation of some findings and future research opportunities. In the discussion, one can highlight why these findings are clinically important, how one can implement the results, reflect on ethical issues and biases from this study. It is also an opportunity to set this study in a wider context, where you can include research from the field of intervention, prevention as well as research related to youth/young adults' use of prescribed opioids/medicine or legal/illegal substance use.

In addition:

·         There is no conclusion in this paper

·         SAWAP is used several times in the article, but the abbreviation is not inserted in the article where the full text name is written the first time.

·         Review the use of commas and periods, some mistakes here and there.

Reviewer 2 Report

I found this added to the understanding of the matrix of issues relevant to teen athletes, injury, and opioid use and a formative approach to addressing the issues using media to engage the participants. The authors offer a practical solution to addressing a potentially dangerous situation. Unfortunately, we all don't have access to a professional production team.

Author Response

Response to Reviewer 1 Comments

Point 1. Where is this study conducted (Europe/Asia/Africa/America)? It would be nice to know the context.

Response 1: We added this detail in the abstract (line 15) in several places in the introduction.

Point 2. In general, there is a lack of literature on the extent of use of pain relievers/prescription opioids in sports internationally for this age group.

Response 2: This is an excellent critique. Although this study is primarily focused on athletes in the United States, we added information in the literature review to reflect that this problem exists throughout the developing world and is in fact seen as a “pandemic”, with the potential to impact athletes everywhere.

Point 3. In addition, I lack background literature related to interventions for youth/young adults. Such information will strengthen this article and provide increased insight and the papers discussion would be improved.

Response 3: Great point. The research that we have cited has also focused on youth/young adults, so we simply emphasized that age group a little more in the literature review.

Point 4: Page 1, line 35, the authors write: "according to a recent national study", which nation?

Response 4: We specified that the study was conducted in the United States.

Point 5: Page 2, line 43, “in only some sports” – can you give examples of the type of sport this applies to?

Response 5: We specified that “some sorts” references spots such as hockey or varsity-level sports among male athletes.

Point 6: Page 2, line 86: is this the correct page reference in the article to Miller-Day & Hecht?

Response 6: Thank you for correcting that error. The original document was paginated in the Health Communication journal as spanning from pages 657 – 670, so our pagination (p. 658) was logical; however, the online article is paginated from pages 1 – 22. It seems less distracting to use the latter pagination scheme, so we revised the in-text citation accordingly. It’s currently referencing page 2 of Miller-Day and Hecht’s article.  

Point 7: Inadequate explanation of the rationale for how the testing of single-group design

Response 7: Good critique. We provided an explanation that for a preliminary feasibility study, a single-group design is acceptable. We also provided a citation of a similar-type study (a feasibility study testing the usability of an online program about concussion management and avoidance among athletes) that used a similar design, to illustrate that there’s precedent to employing this design within the scope of our explorations.

Point 8: Methods: There is a lack of information about how the participants were selected for interview - why these 35 athletes? On what basis were they selected, and not just stating they were selected due to meeting the inclusion criteria.

Response 8: We added information about how those participants were selecting, explaining that a fairly wide net was cast across the United States to recruit participants from a range of school sizes (large and small), demographics (rural and urban), and from a range of school-based sports afforded to them.

Point 9: Who did the interviews, and what kind of interview training did they have?

Response 9: We added this information, explaining that four individuals conducted the interviews. Two of the individuals are professors who teach qualitative methods, and the other two individuals are graduate students. All have been extensively trained in and have experience from other projects with qualitative research, including qualitative data collection and analysis.

Point 10: What was the background for having a screening survey when the participants had to meet the inclusion criteria to participate? It seems unnecessary to have a screening survey just to determine the inclusion/exclusion criteria.

Response 10: We clarified the use of this screening survey, which was used to confirm not only the inclusion criteria but also their consent to be interviewed.

Point 11: There is no information about consent to an interview, how was that handled?

Response 11: Please see Response 10.

Point 12: Are children under the age of 18 competent to consent, or do parents have to consent to children's study participation? This issue is not discussed in the article.

Response 12: We discussed with our IRB the risks and benefits of waiving parental consent for athletes who wanted to participate in our study but did not want their parents to know about their history of misusing prescription opioids. The IRB panel approved a plan to waive parental consent to bolster our sample size, holding that the benefits of interviewing and learning from the higher-risk athletes (i.e., those who had more troubling patterns of opioid misuse) was more important than requiring parental consent and ostensibly “shutting out” those athletes from even considering participation in our study. To protect participants’ privacy we made it explicitly clear in the consent form and in the interviews that no identifying information of the student athletes would be shared with parents.

Point 13: Page 3, line 120-127: this is results and ought to be moved to the result section.

Response 13: Done.

Point 14: Page 3, line 135: who is "the trained interview team"?

Response 14: Our changes in response to Point 9 should not clarify who this trained interview team is referencing. 

Point 15: According to the paper, interviews lasted from 28 to 72 minutes. What was the average time for the interviews? Moreover, can one get in-depth information in a 28 min interview? Bringing this into the discussion as an ethical consideration will strengthen the article.

Response 15: Very good point. That 28-minute interview was one of our first interviews, and its brevity prompted a team-based meeting to brainstorm more follow-up probes and prompts to elicit greater detailed responses among participants. We’ve added that detail to the manuscript, plus the average interview time (52 minutes) across the entire sample.

Point 16: To be able to say something about this study's validity, there is a need for more information about the analysis process. I find it a bit superficial and would like more information here:

o   Was it only one person who did all the coding? (See page 4, line 153).

o   What background did this person have (and the team as well), and how did this background influence the coders perspective/views? Ethical and reflective considerations here would be nice.

o   How did you ensure an 80% inter-code agreement? (see page 4, line164)

o   Who participated in the team (page 4, line 163)? Only the writers or did other participate too?

o   How did you identify themes?

Response 16: We added much more detail to this section, explaining that, although one person (one of the interviewers) did an initial the first “pass” through the transcripts to identify common themes, data analysis was largely a group effort, one that involved pairs of coders to analyze the same transcript, and then to discuss shortly after the coding to look over places of disagreement. Our 80% inter-rater reliability was assessed based on the number of transcript segments that were similarly coded across dyads. 

Point 17: Page 4, line 171: “the following themes emerged….” here you might present all the topics presented in the result section.

Response 17: Done.

Point 18: Can the themes emerge? In an analysis process, the themes do not "appear by themselves", but there are codes/themes that the researchers themselves choose - this is a choice based on preferences, knowledge etc. This is also something that can be dealt with in the method section, and which will improve the article.

Response 18: That’s a really good point. As qualitative researchers we play an active role in interrogating the data, so to suggest that the “themes emerge” hints at a more passive relationship with the transcripts. We’ve made edits so that this deliberate and active naming and choosing is more explicit.   

Point 19: Nice to use quotations in the text, but it would have been better if they were in italics so that they were more visible - and be careful that you use the "-" sign before and after all the quotations. Missing quotation marks on page 5.

Response 19: Quotations have been italicized and quotation marks have been provided for all athletes’ utterances.

Point 20: Page 6, line 250, «Percocet» - why this particular medication? Perhaps an explanation of which preparation this is, as it might not be internationally known.

Response 20: We edited this sentence so that the brand of opioid—Percocet—was replaced with its more generic description of the opioid.

Point 21: Ways to stay healthy, page 6 line 259-280. This part needs more processing. It is not intuitive how these quotes illustrate the themes. The quote on page 6, line 265, how does this illustrate that the mother is an ally when the mother apparently has control over the number of pills?

Response 21: What a good and important catch. Thanks for that. We swapped out one elicitation for another that, we thought, better illustrated “mother as ally.” We also provided some explanation after the transcripts about how those elicitations were not suggesting a controlling relationship between the athletes and their mothers; rather, their mothers gave them a sense of security and freedom in not having to “deal with” the stressors of storing their medicine, taking their medicine, and properly disposing of it when their pain was under control.

Point 22: Summary, line 282-285: I am not sure if the topics (challenges of being an athlete, manageable vs. unmanageable pain, and ways to stay healthy) gives us so much in-depth insight into how they manage/handle their pain and stay healthy. However, the results says something about how they experience the pressure of expectations, how pain is experienced and how the pain affects their life. Do you have more data on what the participants specifically do to manage the pain (besides ending prescribed use earlier or possibly go beyond the prescribed dose/period)? Do they train alternatively/other forms of training? What do they do when they cannot sleep due to the pain? How do the young people/young adults understand being healthy in a context, you are studying? Alternatively, how do they stay healthy when using opioids beyond the prescribed dose/time? Because this says something about what they actually do and may increase our knowledge. As I see it, answers to such questions will provide more in-depth understanding about your study first aim.

Response 22: Good critiques, thank you. We added a bit more information about pain management from the athletes’ perspectives. 

Point 23: Page 8, line 324-325, “five student athletes aged 19 to 22 and one athletic trainer” – how are these participants selected, and why them? Is it completely random or are there some underlying selection criteria?  I miss an explanation of these aspects.

Response 23: We added information on why these individuals were asked to review the prototype.

Point 24: Page 8, line 352, I am not sure if you have enough coverage in the data to say it's a “hyper-awareness” of how to take care of their body. If so, it should be better substantiated.

Response 24: Good point. We removed the descriptor of “hyper.”

Point 25: Page 9, line 359: The abbreviation (SAWP) should appear earlier in the article, first used in line 71.

Response 25: Done.

Point 26: Page 9, line 377: “In addition, the program was shortened” - why, and what was removed?

Response 26: We added an explanation here.

Point 27: Page 9m line 385-394: This is results, and should be moved to the result section.

Response 27: Done.

Point 28: Page 9, line 400-401: “higher score were coded to reflect greater engagement”. How is it possible to gain a higher score than 5 on a 5-step agree-disagree scale?

Response 28: We clarified that higher scores on a scale of 1-5 reflected greater engagement; that original phrasing was indeed confusing, but it was not meant to signify that there were scores that reached higher than 5.

Point 29: Immediate posttest, what does this mean? Is the test taken directly after the test, within a few hours after or on the same day/week as the test was completed?

Response 29: On line 431 and 432 (about a paragraph before the reference you’ve made here), we explain that the posttest is immediately after the intervention.

Point 30: Page 10, line 406: “pre and posttest”, what is the time window for the pre and posttests?

Response 30: We explain that the posttest is immediately after the intervention.

Point 31: Page 10, line 406: “two item adapted from previous research” - which item this is and why are they included? A reference to previous research where this is mentioned must be included.

Response 31: We made clarifications that our use of this item was from past research on the self-efficacy of resisting substance use offers in general; we adapted it so that would measure self-efficacy of resisting opioid prescription misuse.

Point 32: Page 20, line 414: “ORRS” - what is this abbreviation?

Response 32: Thank you for catching this error! This was an acronym for another intervention we developed. We corrected it so that it clearly references SAWP.

Point 33: It is a bit unclear why you present different scores as to whether they were lower or higher than the mean without this being linked to relevance for the intervention. What are you actually measuring and what do you want to show, it is a bit unclear for me.

Response 33: The decision was made to consider the mean score on all scales as an adequate reference point for whether the intervention was associated with increased or decreased measures, posttest. We’re aware that this reference point may seem unclear, but for the purposes of these preliminary analyses, and because our primary intent was to explore the usability of this intervention, the decision to use the mean as a reference point felt appropriate. 

Point 34: Discussion: As a reader, I cannot see that this is a discussion; it is more a form of a brief presentation of some findings and future research opportunities. In the discussion, one can highlight why these findings are clinically important, how one can implement the results, reflect on ethical issues and biases from this study. It is also an opportunity to set this study in a wider context, where you can include research from the field of intervention, prevention as well as research related to youth/young adults' use of prescribed opioids/medicine or legal/illegal substance use.

Response 34: We retained the Discussion heading on line 503, which is a more straightforward discussion about the feasibility trial findings. See Response 35 for more reflections on a conclusion.

Point 35:  There is no conclusion in this paper

Response 35: We reshaped what was originally a second Discussion section into a conclusion where we explain why and how this study is clinically important, how it is novel in its illustration of how an evidence-based, narrative-informed intervention was developed and then tested, and the areas necessary for continued growth in the field. 

Point 36: SAWAP is used several times in the article, but the abbreviation is not inserted in the article where the full text name is written the first time.

Response 36: Fixed. This was a typo and should be SAWP. We made sure that this acronym was not used on its own before the full text was presented.

Point 37: Review the use of commas and periods, some mistakes here and there.

Response 37: Thank you for your thorough review. We edited the entire manuscript for grammar, punctuation, and spelling.